# Is long-bout sedentary behaviour associated with long-term glucose levels 3 months after acute ischaemic stroke? A prospective observational cohort study

Katinka Nordheim Alme [1,2,3] Anne-Brita Knapskog,[4] Halvor Næss,[1,5,6] Mala Naik,[2,7] Mona Beyer,[8,9] Hanne Ellekjaer,[10,11] Coralie English [12,13] Hege Ihle Hansen,[8,14] Camilla Sollesnes Kummeneje,[10] Ragnhild Munthe-Kaas,[8,15] Ingvild Saltvedt,[10,16] Yngve Seljeseth,[17] Xiangchung Tan,[10] Pernille Thingstad,[10] Torunn Askim[10]

For numbered affiliations see end of article.

**Correspondence to**
MD Katinka Nordheim Alme; katinka.alme@gmail.com

## ABSTRACT

**Background and purpose** Sedentary behaviour is a risk factor for vascular disease and stroke patients are more sedentary than their age-matched peers. The association with glucose levels, as a potential mediator, is unclear, and we have investigated the association between long-bout sedentary behaviour and long-term glucose levels in stroke survivors.

**Methods** This study uses data from the Norwegian Cognitive Impairment After Stroke study, a multicentre cohort study. The patients were recruited at hospital admission for acute stroke, and the follow-up was done at the outpatient clinic. Sedentary behaviour—being in a sitting or reclining position—was registered 3 months after stroke using position transition data from the body-worn sensor activPAL attached to the unaffected thigh. A MATLAB script was developed to extract activity data from 08:00 to 10:00 for 4 days and to categorise the data into four bout-length categories. The primary outcome was glycated haemoglobin (HbA1c), analysed at 3 months. Regression models were used to analyse the association between HbA1c and sedentary behaviour in the whole population and stratified based on a diagnosis of diabetes mellitus (DM). Age, body mass index and the use of antidiabetic drugs were added as covariates into the models.

**Results** From a total of 815 included patients, 379 patients fulfilled the inclusion criteria for this study. We found no association between time in sedentary behaviour and HbA1c in the whole stroke population. We found time in sedentary behaviour in bouts of ≥90 min to be associated with a higher HbA1c in patients with DM.

**Conclusion** Long-bout sedentary time is associated with a higher HbA1c in patients with DM 3 months after ischaemic stroke. Future research should investigate the benefit of breaking up sedentary time as a secondary preventive measure.

**Trial registration number** NCT02650531, https://clinicaltrials.gov/ct2/show/NCT02650531

## Strengths and limitations of this study

► The study investigates the association between long-bout sedentary behaviour and long-term glucose levels in a large cohort.
► Sedentary behaviour is measured objectively, includes information about bout lenght and the method is valid for the stroke population.
► The outcome is a valid measure for long-term glucose levels.
► We have included information about diabetes mellitus, body mass index and medication use.
► Information about diet, details of medication use and the type of diabetes mellitus including level of insuline resitance would have increased the explanatory abilities of the model, but were not accessible.

## INTRODUCTION

Sedentary behaviour is associated with negative health outcomes, especially when accumulated in long bouts and in the least active individuals. Hence, introducing 'sedentary breaks' as an interventional measure has gained interest.[1–3] Sedentary behaviour is associated with vascular disease, like stroke, presumably through metabolic and inflammatory pathways.[4–7] Stroke patients are more sedentary than their age-matched peers from the general population,[8] and targeting sedentary behaviour as a secondary preventive strategy after stroke is recommended.[9] However, details about how different patterns of sedentary behaviour relate to vascular risk factors in different populations are not entirely understood and need to be more thoroughly explored.[9 10]

One unresolved issue is the relationship between sedentary behaviour and glucose metabolism.[4 5 11–14] The relationship is

complex and is dependent on the characteristics of the target population with respect to food intake, habitual activity level, age, body composition, dominating muscle fibre type, diseases (such as diabetes mellitus (DM)) and medication use.[2 4 15–18] Sample size, choice of methods of measuring and analysing sedentary time, and how and when to measure markers of glucose metabolism vary between studies and make it difficult to synthesise the available evidence.[4 11–14 19] Also, many prior studies have relied on questionnaire-based measures of sedentary behaviour.[13] More recently, the use of accelerometer-based technology is increasing, and this allows for more reliable and detailed information about sedentary behaviour, as opposed to self-reported activity.[13]

In this study, we have investigated habitual sedentary behaviour and the association to long-term glucose levels, measured by glycated haemoglobin (HbA1c), in a stroke population, using objective measures of activity. The primary aim was to investigate the association between long-bout sedentary behaviour and glucose levels in stroke survivors. The secondary aim was to investigate how this association was altered in the presence of prestroke DM.

Our hypotheses were that sedentary behaviour in long bouts was associated with long-term glucose levels in an unselected stroke population and that the association would be more pronounced in the subgroup of patients suffering from DM.

## MATERIALS AND METHODS

The patients were part of the Norwegian Cognitive Impairment After Stroke (Nor-COAST) study, a prospective cohort study recruiting acute stroke patients from five contributing hospitals from May 2015 to March 2017.[20] The patients were assessed at hospital admission and after 3, 18 and 36 months. Inclusion criteria were (1) acute stroke, according to the WHO criteria, arriving at hospital within 1 week after symptom onset; (2) above 18 years of age; (3) ability to understand Norwegian; and (4) ability to give informed consent. For patients unable to provide consent for themselves, the next of kin may give oral consent. Exclusion criteria were (1) not living in the catchment area of one of the inclusion hospitals, (2) the symptoms explained by other diagnosis than stroke, (3) short life expectancy (<3 months) or a modified Rankin Scale (mRS) score of 5, except for patients included at the main centre, St. Olavs Hospital.

Some additional criteria were made for this substudy: patients were included only if they attended the 3-month follow-up, had ischaemic stroke (including those with haemorrhagic transformation), were able to walk 50 m with a walker or personal support (Barthel Index (BI) item 9: ≥10 points), had blood samples taken and valid activity data, all at 3 months.

Assessment at baseline was performed on day 7 after symptom onset or at discharge from hospital if this occurred earlier. The assessments were performed by trained research assistants at the outpatient clinic, using a standardised case report form.[20]

Sedentary behaviour was measured at 3 months by registering position transition with a single thigh-worn sensor (activPAL3, Model 20.2; PAL Technologies, Glasgow, UK) on the unaffected thigh for seven consecutive days. Only patients with recordings from at least four full days were included. Activity was analysed during daytime, defined as between 08:00 am and 10:00 pm. Sedentary events were divided if they crossed these time boundaries. Manual inspection of the output to identify non-wear time was performed. Sedentary behaviour was defined as sitting or lying. The threshold for noise was 1.5 s and sedentary events were merged if they were broken by events of standing of ≤3 s.

A custom-made MATLAB script (V.R2016b Math-Works, Natick, Massachusetts, USA) was developed to extract frequency and duration of sedentary bouts into predefined bout categories (see Statistics section).

Stroke severity was measured by the National Institutes of Health Stroke Scale on admission and at the three month follow-up, global function by the mRS[21] and basic activities of daily living by the BI.[22] Non-fasting blood samples were analysed for HbA1c %. Body mass index (BMI, kg/m$^2$) was calculated from height and weight. Medications were analysed based on the Anatomical Therapeutic Chemical (ATC) classification system. The diagnosis of DM was defined at baseline by medical history or medication use (ATC: A10) or finding of HbA1c≥6.5% at baseline. Hypercholesterolaemia at baseline was defined by medication use (ATC: C10) or total cholesterol of >6.2 mmol/L or/and low-density lipoprotein of ≥4.1 mmol/L at baseline. Hypertension at baseline was based on medication use (ATC: C02, C03, C04, C07, C08 and C09) on admission.

## Statistics

Differences between patients who were included and not included, and between those with and without DM with respect to characteristics at baseline and 3-month follow-up and sedentary behaviour were analysed using t-test and χ$^2$ test. The results are shown as frequency and per cent or mean and SD. The use of the t-test was based on the central limit theorem.

Sedentary behaviour is displayed as daily averages (hours/day) of total sedentary time and number of bouts by bout length (<30, 30–59, 60–89 and ≥90 min).

A linear regression was used to analyse the association between the dependent variable, HbA1c and the independent variables total sedentary time, sedentary time at different bout lengths, BMI, age and the use of antidiabetic drugs. A multiple linear regression model was made using the same dependent and independent variables, except total sedentary time because of collinearity. The model was tested for multicollinearity. The covariates were added by forced entry based on literature.[17] The analysis was done using the whole population and stratified based on the presence of a diagnosis of DM, because

of the alterations in glucose metabolism in the patients with DM.[17] A standardised regression coefficient, CI and p value are presented. The residuals of the regressions were not normally distributed; thus, for the significance test, we used a cubic transformed version of the dependent variable, giving a normal distribution of the residuals of the regressions.

Missing data were not imputed as this was limited to 10.8% of the population (HbA1c missing in 4.5% of the patients and in BMI for 6.9% of the patients).

The significance level was set to 0.05, but since we have not made any formal correction for multiple comparison, p values above 0.01 should be interpreted with caution.

The power calculation was made for the main study. For this study, we made a post hoc power calculation for the stratified multiple regression. For the smallest group, n=70, $R^2$=0.37; probability level was 0.05; and calculated beta was 0.99.

Collinearity was checked using the Pearson product–moment correlation coefficient with a cut-off of ≥0.6. The significance level was set to p<0.05. Multicollinearity was checked by investigating the variance inflation factor (VIF), with a tolerance level of 1/VIF>0.1.

The statistical analyses were conducted in Stata/SE V.16.0 for Windows, revision 01 Aug 2019.

### Patient and public involvement

The Nor-COAST study has included one stroke patient and three spouses representing the national unions for patients with stroke and dementia. The user representatives have been actively participating in the planning and performance of the Nor-COAST study, including the choice of analytical approach and the dissemination of results to the users. They have been invited to meetings for the Nor-COAST research group, and we have held separate meetings with them two to three times per year where substudies, such as this study, are presented.

### RESULTS

A total of 815 patients were included in the study, 700 assessed at 3 months, 636 who had ischaemic stroke. Of these, 379 fulfilled the inclusion criteria for this study (figure 1).

Details are shown in table 1. There were 218 men (57.8%); the mean age was 71.5 (SD 11.4) years, and they had mean BI scores of 90.9 (SD 15.8) and 97.5 (SD 6.7) points at baseline and 3 months, respectively.

At baseline, a diagnosis of DM was registered in 74 (19.5%) patients, and these patients had a higher mean BMI (27.6 kg/m² (SD 4.3) vs 25.8 kg/m² (SD 4.3), p=0.002) and more often had hypercholesterolaemia (73.0% vs 50.8%, p=0.001) and hypertension (79.7% vs 48.5%, p<0.001). The groups were otherwise similar at baseline. At 3 months, patients with DM still had a higher mean BMI (28.3 kg/m² (SD 4.7) vs 26.3 kg/m² (SD 4.3), p=0.001) and HbA1c (7.0% (SD

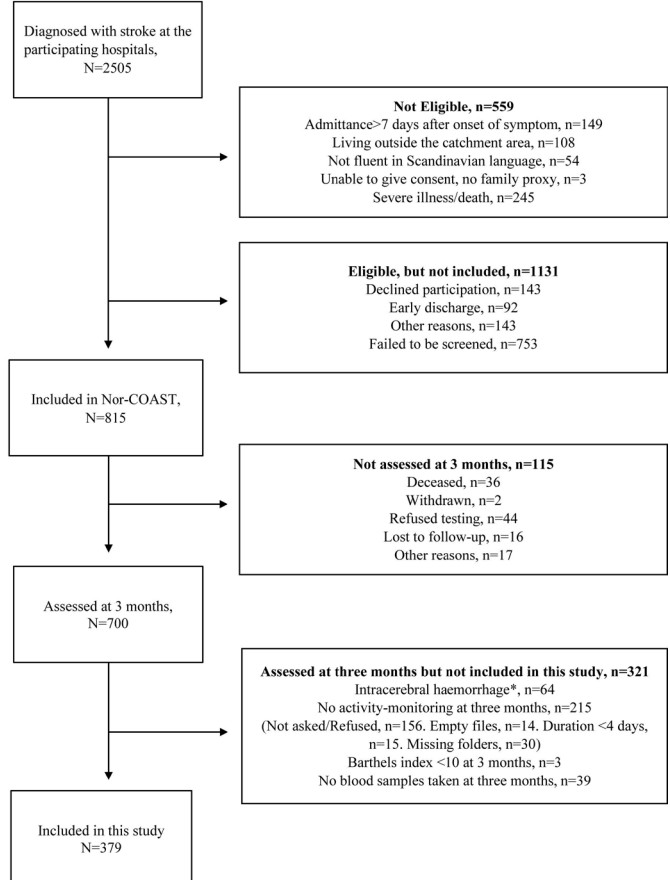

**Figure 1** Flowchart for inclusion to the Norwegian Cognitive Impairment After Stroke-study (Nor-COAST).*Not haemorrhagic transformation.

1.6) vs 5.6% (SD 0.4), p<0.001). Of the patients with DM, 47 (63.5%) used antidiabetic drugs, and of these, 16 (21.6%) used insulin.

The distribution of sedentary behaviour is presented in table 2 and figure 2. Patients with DM spent more time (mean hours/day) in sedentary behaviour (10.2 (SD 1.8) vs 9.6 (SD 1.2), p=0.008), more time (mean hours/day) in sedentary bouts of 30–59 min (2.7 (SD 0.9) vs 2.4 (SD 0.9), p=0.001) and a higher number of bouts of 30–59 min (3.8 (SD 1.2) vs 3.4 (SD 1.2), p=0.017) and ≥90 min (1.0 (SD 0.7) vs 0.8 (SD .7), p=0.026) per day, respectively.

The results from the unadjusted regressions are found in table 3 and figure 3. The adjusted analysis is found in table 4. There was a significant association between total sedentary time and sedentary time in bouts lasting 60–89 min and ≥90 min and HbA1c in the whole population. It was also a significant association between total sedentary time and sedentary time in bouts of ≥90 min in the subgroup of patients with DM in the unadjusted analysis. In the adjusted analyses, sedentary time in bouts of ≥90 min (β=0.43, CI 0.17 to 0.72, p=0.008) was associated with a higher HbA1c in patients with DM. The model explained 38% of the variance in HbA1c ($R^2$=0.38).

**Table 1** Baseline and 3 months characteristics

| Characteristics | Not included (n=321) | Included (n=379) | P value | No DM (n=305) | DM (n=74) | P value |
|---|---|---|---|---|---|---|
| Gender, male, n (%) | 181 (56.4) | 218 (57.8) | 0.710 | 174 (57.1) | 45 (60.8) | 0.557 |
| Age at baseline (years), mean (SD) | 74.0 (12.2) | 71.5 (11.4) | 0.005 | 71.3 (11.8) | 72.0 (9.4) | 0.625 |
| At 3 months | 74.8 (12.2) | 72.3 (11.4) | 0.004 | 72.1 (11.8) | 72.8 (9.3) | 0.656 |
| BMI (kg/m$^2$), mean (SD) | 25.6 (4.0) | 26.2 (4.4) | 0.101 | 25.8 (4.3) | 27.6 (4.3) | 0.002 |
| At 3 months | 26.4 (3.8) | 26.7 (4.4) | 0.370 | 26.3 (4.3) | 28.3 (4.7) | 0.001 |
| NIHSS score on admission, mean (SD) | 4.8 (6.0) | 3.4 (4.2) | <0.001 | 3.4 (4.3) | 3.4 (4.0) | 0.981 |
| At 3 months | 1.3 (2.3) | 0.7 (1.3) | <0.001 | 0.7 (1.3) | 0.6 (1.1) | 0.579 |
| mRS, mean (SD) | 2.5 (1.4) | 1.9 (1.2) | <0.001 | 2.0 (1.3) | 1.9 (1.1) | 0.728 |
| At 3 months | 2.2 (1.5) | 1.4 (0.9) | 0.000 | 1.4 (0.5) | 1.4 (0.9) | 0.448 |
| mRS score ≤2, n (%) | 154 (48.1) | 262 (69.5) | <0.001 | 207 (68.1) | 55 (74.3) | 0.297 |
| At 3 months | 194 (60.4) | 334 (88.4) | <0.001 | 267 (87.5) | 67 (91.8) | 0.310 |
| BI, mean (SD) | 79.4 (26.6) | 90.9 (15.8) | <0.001 | 90.3 (16.4) | 93.4 (12.5) | 0.130 |
| At 3 months | 86.8 (22.9) | 97.5 (6.7) | <0.001 | 97.5 (7.1) | 97.4 (5.1) | 0.908 |
| BI score ≥95, n (%) | 165 (51.4) | 260 (68.6) | <0.001 | 203 (66.6) | 57 (77.0) | 0.082 |
| At 3 months | 211 (65.7) | 341 (90.0) | <0.001 | 275 (90.5) | 66 (89.2) | 0.802 |
| BI item 9≥10 points at baseline, n (%) | 273 (85.3) | 366 (96.6) | <0.001 | 293 (96.1) | 73 (98.7) | 0.598 |
| At 3 months | 292 (91.0) | 379 (100.0) | <0.001 | – | – | – |
| Living conditions before stroke, n (%) | | | 0.000 | | | 0.427 |
| At home | 314 (97.8) | 379 (100) | | 304 (99.7) | 74 (100) | |
| Without home nursing care | 261 (81.3) | 358 (94.5) | | 290 (95.1) | 68 (91.9) | |
| With home nursing care | 49 (15.3) | 20 (5.3) | | 14 (4.6) | 6 (8.1) | |
| Residential care* | 4 (1.3) | 1 (0.3) | | 1 (0.3) | 0 | |
| Nursing home | 7 (2.2) | 0 | | 0 | 0 | |
| DM at baseline, n (%) | 54 (16.8) | 74 (19.5) | 0.357 | – | – | – |
| HbA1c %, mean (SD) | | | | | | |
| At 3 months, mean (SD) | 5.7 (0.8) | 5.9 (0.8) | 0.102 | 5.6 (0.4) | 7.0 (1.6) | <0.001 |
| Antidiabetic drugs, (%) | | | | | | |
| All at 3 months, n (%) | 35 (10.9) | 47 (12.4) | 0.529 | 0 | 47 (63.5) | <0.001 |
| Insulin use at 3 months, n (%) | 15 (4.7) | 16 (4.2) | 0.772 | 0 | 16 (21.6) | <0.001 |
| Hypercholesterolaemia baseline, n (%) | 145 (45.2) | 209 (55.2) | 0.009 | 155 (50.8) | 54 (73.0) | 0.001 |
| Hypertension baseline, n (%) | 175 (54.5) | 207 (54.6) | 0.979 | 148 (48.5) | 59 (79.7) | <0.001 |
| Prior cerebrovascular disease, n (%) | 84 (26.2) | 80 (21.1) | 0.115 | 64 (21.0) | 16 (21.6) | 0.904 |

BMI=weight/height$^2$.
*Residental care: own customised apartment with home nursing care.
BI, Barthel Index; BMI, body mass index; DM, diabetes mellitus; HbA1C, glycated haemoglobin; mRS, modified Rankin Scale; NIHSS, National Institute of Health Stroke Scale.

For the other covariates in the unadjusted analyses (table 3), BMI and antidiabetic drugs were seen to be significantly associated with a higher HbA1c in the whole population. BMI and age were associated with HbA1c in the non-diabetic group, while in the group of patients with DM, only antidiabetic drugs were associated with HbA1c. In the adjusted analyses (table 4), we found the same pattern except that the BMI was not being significantly associated with HbA1c in the whole population.

## DISCUSSION

The primary aim of this study was to investigate the association between long-bout sedentary behaviour and long-term glucose levels, measured by the HbA1c in stroke survivors. We found no significant associations. The secondary aim was to investigate any differences in the association between the patients with or without DM. We found sedentary behaviour in bouts of ≥90 min to be associated with a higher HbA1c in patients with DM.

**Table 2** Mean sedentary time per day (hours) total and by bout length category 3 months after stroke

| | Hours/day, mean (SD) | | | Bouts/day (n), mean (SD) | | |
|---|---|---|---|---|---|---|
| | No DM | DM | P value | No DM | DM | P value |
| Total sedentary time | 9.6 (1.2) | 10.2 (1.8) | 0.008 | 43.3 (14.3) | 40.4 (12.0) | 0.107 |
| Bout-length category (min) | | | | | | |
| <30 | 4.0 (1.2) | 3.7 (1.0) | 0.097 | 37.8 (15.0) | 34.2 (12.8) | 0.056 |
| 30–59 | 2.4 (0.9) | 2.7 (0.9) | 0.001 | 3.4 (1.2) | 3.8 (1.2) | 0.017 |
| 60–89 | 1.4 (0.9) | 1.6 (1.0) | 0.118 | 1.2 (0.7) | 1.3 (0.8) | 0.142 |
| ≥90 | 1.8 (1.7) | 2.2 (1.6) | 0.050 | 0.8 (0.7) | 1.0 (0.7) | 0.026 |

Mean daytime sedentary behaviour per day over a period of 4 days.
Daytime: 08:00–10:00.
DM, diabetes mellitus.

Results from studies investigating the associations between level of activity and glucose metabolism in general have varied.[5 11 12 14] For stroke in particular, Moore *et al* found an association between energy expenditure and glucose and insulin sensitivity (HOMA), not adjusted for BMI, age, DM or medication use.[23] Beyond that, high quality studies on the impact of sedentary behaviour in the stroke population are scarce. Much of the discrepancies seen in the literature can be explained by differences in applied methodology. We will therefore make an outline of the results in the context of (1) study population, (2) methods for measuring and analysing sedentary behaviour and (3) methods for measuring glucose metabolism.

The population in this study are people with ischaemic stroke who are relatively old. We also have a subgroup of patients with DM. All these factors are relevant in the context of sedentary behaviour and glucose metabolism. As expected from our knowledge of sedentary behaviour patterns in stroke patients, we find that our patients are more sedentary than their age-matched peers from the general population.[8 24] The negative impact of sedentary behaviour is found to be stronger for the most sedentary.[2] It is also important to keep in mind the age related changes in glucose metabolism and altered glucose metabolism due to DM when interpreting our results.[17 25] Physical activity is known to increase both the transcription of and the translocation of glucose transporter type 4 (GLUT-4), responsible for the transportation of glucose into muscle and fat cells. Contraction-stimulated GLUT-4 reallocation can, in part, counter-act the down-regulation seen in patients with reduced insulin sensitivity, such as patients with diabetes type 2.[25 26] Hence, the association between sedentary behaviour and a higher HbA1c in the participants with DM might be partly explained by these mechanisms.[26 27]

Historically, sedentary behaviour has been measured and analysed using a variety of tools and methods.[1 5 11 13 28] In this study, we used time in sedentary behaviour in a given bout-length category, as recommended by the Sedentary Behaviour Research Network (SBRN).[13] The combined effect of time and bout length gives a more nuanced measure of the exposure compared with mere bout frequency, number of breaks or mean bout duration.

In a consensus guideline, the SBRN in 2017 presented a phenomenological definition of sedentary behaviour as 'any waking behaviour characterised by an energy expenditure ≤1.5 metabolic equivalents (METs) while in a sitting or reclining posture'.[13] How to measure energy expenditure is not defined, but body-worn sensors are preferred.[13] One common method is to convert accelerometer counts into metabolic equivalents (METs).[13] This conversion is based on healthy norms and the method

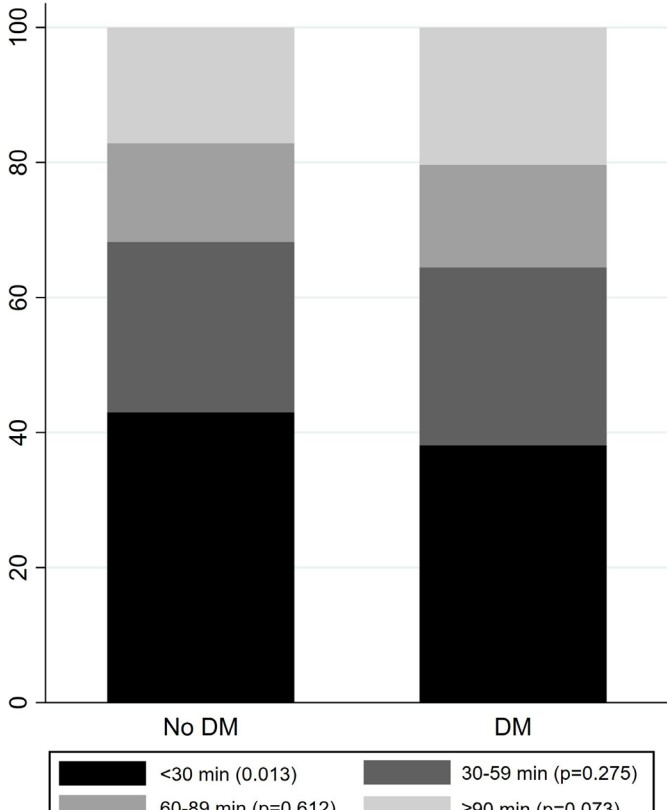

**Figure 2** Differences in accumulation patterns of sedentary behaviour by bout length categories in per cent of mean sedentary time between patients with or without DM. DM, diabetes mellitus.

Legend for Figure 2:
- <30 min (0.013)
- 30-59 min (p=0.275)
- 60-89 min (p=0.612)
- ≥90 min (p=0.073)

**Table 3** Unadjusted linear regressions investigating the associations between HbA1c and sedentary time*, BMI, age and use of antidiabetic drugs in patients with or without DM

| | All patients (n=379) | | | No DM (n=305) | | | DM (n=74) | | |
|---|---|---|---|---|---|---|---|---|---|
| | β | CI | P value | β | CI | P value | β | CI | P value |
| Total sedentary time | 0.18 | 0.08 to 0.29 | <0.001 | −0.03 | −0.15 to 0.08 | 0.598 | 0.38 | 0.16 to 0.60 | 0.001 |
| Bout-length category (min) | | | | | | | | | |
| <30 | −0.07 | −0.20 to 0.00 | 0.059 | −0.06 | −0.18 to 0.05 | 0.273 | −0.08 | −0.31 to 0.15 | 0.510 |
| 30–59 | 0.09 | −0.01 to 0.20 | 0.075 | 0.03 | −0.09 to 0.14 | 0.616 | 0.04 | −0.20 to 0.27 | 0.754 |
| 60–89 | 0.14 | 0.04 to 0.25 | 0.005 | 0.06 | −0.06 to 0.17 | 0.326 | 0.24 | −0.02 to 0.44 | 0.067 |
| ≥90 | 0.07 | −0.03 to 0.17 | 0.007 | −0.03 | −0.15 to 0.08 | 0.595 | 0.32 | 0.10 to 0.54 | 0.006 |
| BMI | 0.15 | 0.04 to 0.25 | 0.002 | 0.16 | 0.04 to 0.28 | 0.009 | −0.10 | −0.34 to 0.14 | 0.427 |
| Age | 0.08 | −0.02 to 0.19 | 0.110 | 0.17 | 0.06 to 0.29 | 0.003 | 0.05 | −0.18 to 0.29 | 0.657 |
| Antidiabetic drugs | 0.72 | 0.50 to 0.93 | <0.001 | – | – | – | 0.51 | 0.10 to 0.93 | <0.001 |

BMI=weight/height$^2$.
Antidiabetic drugs are defined as Anatomical Therapeutic Chemical A10.
*Sedentary time is analysed as total time (mean hours/day) and by bout-length category (mean hours/day).
.BMI, body mass index; DM, diabetes mellitus; HbA1c, glycated haemoglobin.

is not validated for the older and frailer population in general. Stroke patients in particular have been shown to have a higher energy expenditure when walking,[29] and there is no valid conversion norm for this patient group. In a healthy population, the energy expenditure of standing has been estimated to be 1.59 METs,[30] and data regarding position change from sitting to standing has shown to be accurate for the stroke population.[31] We have therefore chosen to use sitting and lying position as an approximation for sedentary behaviour.

Finally, methods for measuring glucose metabolism have changed, following the revised diagnostic criteria for DM, towards using HbA1c instead of fasting and 2 hour glucose. In this study we have used HbA1c as it represents the mean glucose level in a 3 month period and is not affected by recent changes in diet or activity. Compared with other measures of glucose metabolism, HbA1c is more convenient in regard to fasting state, with better analytic stability and less day-to-day variation. Any potential difference in regards to predictive value for future vascular disease is not entirely clear.[32] The homeostasis model assessment of insulin resistance could have been a useful supplement, as one might suspect the relative importance of sedentary behaviour on glucose

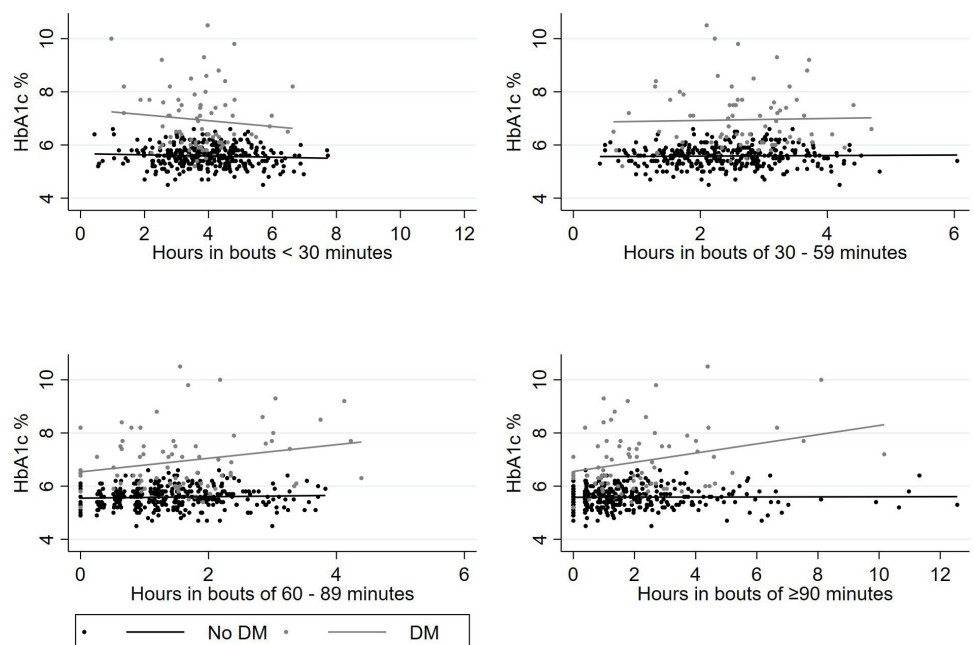

**Figure 3** Association between the amount of sedentary time (hours) and HbA1c (%) by different sedentary time bout-length categories and HbA1c value shown for patients with our without a diagnosis of DM. DM, diabetes mellitus; HbA1c, glycated haemoglobin.

**Table 4** Adjusted relationship between HbA1c and sedentary time by bout-length categories (mean hours/day), BMI, age and use of antidiabetic drugs in patients with or without DM

| | All patients (n=338) | | | No DM (n=270) | | | DM (n=68) | | |
|---|---|---|---|---|---|---|---|---|---|
| | β | CI | Pvalue | β | CI | P value | β | CI | P value |
| Bout-length category (min) | | | | | | | | | |
| <30 | 0.02 | −0.08 to 0.12 | 0.725 | −0.08 | −0.25 to 0.08 | 0.311 | 0.24 | −0.03 to 0.50 | 0.052 |
| 30–59 | 0.01 | −0.06 to 0.09 | 0.734 | 0.01 | −0.13 to 0.12 | 0.910 | 0.06 | −0.15 to 0.27 | 0.292 |
| 60–89 | 0.04 | −0.05 to 0.13 | 0.363 | 0.01 | −0.15 to 0.13 | 0.899 | 0.16 | −0.05 to 0.38 | 0.214 |
| ≥90 | 0.04 | −0.05 to 0.13 | 0.417 | −0.12 | −0.27 to 0.02 | 0.105 | 0.43 | 0.17 to 0.72 | 0.008 |
| BMI | 0.06 | 0.02 to 0.14 | 0.127 | 0.20 | 0.08 to 0.33 | 0.001 | −0.2 | −0.37 to 0.04 | 0.202 |
| Age | 0.05 | −0.03 to 013 | 0.198 | 0.20 | 0.07 to 0.33 | 0.002 | −0.1 | −0.28 to 0.15 | 0.978 |
| Antidiabetic drugs | 0.71 | 0.48 to 0.93 | <0.001 | – | – | – | 0.4 | −0.04 to 0.84 | <0.001 |

BMI=weight/height$^2$.
Antidiabetic drugs are defined as Anatomical Therapeutic Chemical A10.
BMI, body mass index; DM, diabetes mellitus; HbA1C, glycated haemoglobin.

metabolism to be higher in those with insulin resistance than in those with insulin deficiency. This was not included in the laboratory work-up in the study, among other reasons because fasting blood samples were not feasible.[33]

In our study, the diagnosis of hypercholesterolaemia and hypertension is, among others, based on the use of medications. Thus, in line with national guidelines[34] for primary protective strategies in DM, we found a higher rate of patients defined as having hypercholesterolaemia and hypertension in patients with DM.

The aim of this study was to investigate daytime sedentary behaviour. Hence, sleep time, predefined as 10 hours from 10:00 pm until 08:00 am, was excluded from the analysis. A study by Ezeugwu and Manns has shown an average sleep duration of 8.9 hours (range 6.6–11.6) in stroke patients.[35] By making this assumption with regard to sleep patterns, we might have underestimated sedentary behaviour. However, a quality check of the daytime data against the 24-hour data showed that more than 80% of the short sedentary bouts (<30 min) occurred between 08:00 pm and 10:00 am, indicating that we actually have succeeded in capturing daytime activity and excluding sleep time. Nevertheless, future research should focus on developing algorithms that are able to extract sleep time from the 24-hour data in order to capture a greater diversity of activity patterns.

With the exception of one hospital, patients with a mRS score of 5 were not included in the NorCOAST study. In this subsample, patients had to be able to walk 50 m with walking aids or personal support and be fit enough to come to the outpatient clinic. Thus, this population is healthier and fitter than the general stroke population, reducing the generalisability of our results.

### Strengths and limitations
This study has several strengths. It was done on a large sample of stroke patients, and all of the assessments were done at 3 months poststroke. We have objective registrations of sedentary behaviour, reflecting the habitual level of physical activity of the patient. We have considered the contribution of potential confounders, factors associated with glucose metabolism, such as age, medication use, DM and BMI. We have investigated the impact of sedentary behaviour at different bout lengths, hence getting a more nuanced evaluation of sedentary behaviour.

There are some limitations to our study. Diet, details about medication, the relative contribution of insulin deficiency versus insulin resistance and the intensity of physical activity are not accounted for. These factors would be associated with the outcome, but not with the explanatory variable, and hence were not real confounders. Information about these factors would have increased the explanatory abilities of the model, but would not have changed the association.

### SUMMARY
This study did not find an association between sedentary behaviour and HbA1c in a stroke population 3 months after stroke. However, we identified an association between long-bout sedentary behaviour and a higher HbA1c in patients with DM. The results are in agreement with knowledge about glucose consumption in general and in patients with DM in particular. Reducing long-bout sedentary behaviour in patients with DM might be an important target for secondary prevention, but the results need to be verified by experimental studies. If confirmed, this will increase our understanding of the causative pathways between sedentary behaviour and vascular risk.

**Author affiliations**
[1]Department of Clinical Medicine (K1), University of Bergen, Bergen, Norway
[2]Department of Internal Medicine, Haraldsplass Deaconess Hospital, Bergen, Norway
[3]Kavli Research Centre for Geriatrics and Dementia, Haraldsplass Deaconess Hospital, Bergen, Norway
[4]Department of Geriatric Medicine, Oslo University Hospital Ullevaal, Oslo, Norway

[5]Department of Neurology, Haukeland University Hospital, Bergen, Norway
[6]Centre for Age-Related Medicine (SESAM), Stavanger University Hospital, Stavanger, Norway
[7]Department of Clinical Science (K2), University of Bergen, Bergen, Norway
[8]Institute of Clinical Medicine, University of Oslo, Oslo, Norway
[9]Division of Radiology and Nuclear Medicine, Oslo University Hospital, Oslo, Norway
[10]Department of Neuromedicine and Movement Science, Norwegian University of Science and Technology, Trondheim, Norway
[11]Stroke Unit, Clinic of Internal Medicine, Saint Olavs Hospital University Hospital, Trondheim, Norway
[12]Division of Health Sciences, International Centre for Allied health Evidence, University of South Australia Division of Health Sciences, Adelaide, South Australia, Australia
[13]Stroke Division, The Florey Institute of Neuroscience and Mental Health, Parkville, Victoria, Australia
[14]Department of Neurology, Oslo University Hospital, Oslo, Norway
[15]Department of Medicine, Vestre Viken Hospital Trust, Drammen, Norway
[16]Department of Geriatrics, Clinic of Internal Medicine, Saint Olavs Hospital University Hospital, Trondheim, Norway
[17]Department of Internal Medicine, Aalesund Hospital, Alesund, Norway

**Contributors** Project administration: IS. Funding acquisition: IS and KNA. Conceptualisation: KNA, TA, A-BK, HN and MN. Methodology: PT, TA, XT, CSK and CE. Validation: CSK, PT and TA. Project administration: MB, PT, IS, TA, HE, HIH, RM-K and YS. Investigation: HIH, CSK, RM-K, MB, HE, HIH, CSK, RM-K, YS, XT, PT and YS. Data curation: PT, CSK, RM-K and KNA. Software: XT, KNA, CSK and PT. Formal analysis: KNA. Interpretation: KNA, TA and HN. Writing (original draft preparation): KNA. Visualisation: KNA and TA. Writing, reviewing and editing: KNA, A-BK, HA, MN, MB, HE, CE, HIH, CSK, RM-K, IS, YS, XT, PT and TA. Supervision: TA, A-BK, HN and MN. All authors read and agreed to the published version of the manuscript and are accountable for the content.

**Funding** The author has been granted salary from Haraldsplass Deaconess Hospital and a scholarship from the Western regional health authority of Norway to do this work. The Nor-COAST study has been financed by the Norwegian Health Association and the Norwegian University of Science and Technology (grant/award number: not applicable).

**Competing interests** A-BK: principal investigator in two drug trials (Roche BN29553 and Boehringer-Ingelheim 1346-0023) at the memory clinic, Oslo University Hospital. Others: none declared.

**Patient and public involvement** Patients and/or the public were involved in the design, or conduct, or reporting, or dissemination plans of this research. Refer to the Materials and methods section for further details.

**Patient consent for publication** Not required.

**Ethics approval** This study was conducted in accordance with the institutional guidelines and was approved by the Regional Committee of Medical and Health Research Ethics (REK no: 2017/2060/REK Midt, 2015/171/REK Nord). Due to Norwegian regulations and conditions for informed consent, the dataset is not publicly available. The study was registered with Clinicaltrials.gov.

**Provenance and peer review** Not commissioned; externally peer reviewed.

**Data availability statement** No data are available. Due to Norwegian regulations and conditions for informed consent, the dataset is not publicly available.

**ORCID iDs**
Katinka Nordheim Alme http://orcid.org/0000-0002-3925-4169
Coralie English http://orcid.org/0000-0001-5910-7927

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
