## [Reviewer comments · BMJ Open]

ARTICLE DETAILS

TITLE (PROVISIONAL)	Is long-bout sedentary behaviour associated with long term glucose levels three months after acute ischemic stroke? A prospective observational cohort study.
AUTHORS	Alme, Katinka; Knapskog, Anne-Brita; Næss, Halvor; Naik, Mala; Beyer, Mona; Ellekjaer, Hanne; English, Coralie; Hansen, Hege; Kummeneje, Camilla; Munthe-Kaas, Ragnhild; Saltvedt, Ingvild; Seljeseth, Yngve; Tan, Xiangchung; Thingstad, Pernille; Askim, Torunn

VERSION 1 – REVIEW

REVIEWER	N. Jennifer Klinedinst University of Maryland, Baltimore
REVIEW RETURNED	16-Mar-2020

GENERAL COMMENTS	Review of Manuscript Number bmjopen-2020-037475: "Is long bout sedentary behaviour associated with long term glucose levels in stroke patients three months after acute ischemic stroke? A prospective observational cohort study." This article describes a longitudinal observational study of the relationship between sedentary behavior and HbA1c in ischemic stroke patients. This topic is interesting and adds to the literature knowledge of long term bouts of sedentary behavior of glucose metabolism for all stroke patients and then broken down by diagnosis of Diabetes mellitus or not using data available from the Nor-COAST study in Norway. Introduction: The authors provide a sound and logical introduction to the importance of the problem of understanding different types of sedentary behavior and the impact on glucose metabolism in stroke patients. Methods Design: The design is logical and the objective measure of activity using a thigh based sensor (accelerometer) is a particular strength. One thing that should be addressed is they excluded analyses from participants who did not have at least 4 days of sensor data, and these folks may be inherently different than those that did not capture 4 days of data. This should be presented. Analyses were performed on the whole sample and by comparing those with and without diabetes. They did not correct for multiple analyses, but cautioned readers to take conclusions above 0.01 with caution. The statistical analyses are appropriate given the hypotheses. Results: Results were presented clearly and logically. The major
---

	findings were that HbA1c was not related to sedentary behavior in the overall sample, but was for the diabetes group, when compared to the non-diabetes group. Discussion: The first two paragraphs in the discussion are essentially a re-statement of the findings in the results section and other redundancies. The authors compare their findings to the existing literature, but here the writing is less strong and harder to follow their logic. This could be improved by being more of a synthesis of results as compared to existing literature and an integration of how this knowledge could be incorporated into to clinical care of stroke patients. General: This manuscript would benefit from some general editing as there are a few grammatical mistakes and redundancies. It is otherwise a worthy paper, given some editing.
--	---

REVIEWER	Monica Serra UT Health San Antonio, USA
REVIEW RETURNED	20-Mar-2020

GENERAL COMMENTS	This manuscript examined the relationship between bouts of sedentary behavior and long term glucose control (by HbA1c) in stroke patients. The authors find that there is no association in the overall group, but when stratified by diabetes status, there is an association in those with diabetes. There are several clarifying that may aid the interpretation of this manuscript described below. The discussion should a greater description of why this relationship is observed only in those with diabetes and not the whole group other than to say that the evidence is variable considering that most studies support a relationship between sedentary behavior and diabetes risk. Including data from 18 and 36 months would strengthen this paper. Consider removing “in stroke patients to update the tile to “Is long bout sedentary behaviour associated with long term glucose levels three months after acute ischemic stroke? A prospective observational cohort study” as it is already clear from the title that it is in stroke patients. Mechanism of stroke may be different in those <50 vs 50+ years as most strokes occur after the age of 50 years. Please justify the inclusion of those <50 years. What % of the group did these individuals encompass? Please justify the inclusion of only those able to walk >50 m? How many were excluded based upon these criteria? Were nightshift workers included? Did all stroke survivors report that their normal awake time was 8am-10pm? Why include only this range when some individuals may be stay awake later and get up earlier and this would affect their sedentary time calculation. BMI: update unit to kg from mg in the methods What % were on insulin, indicating more severe diabetes? Does this subset respond similarly? Though the discussion attempts to put into context why HbA1c was used, measures of glucose/insulin/HOMA-IR may help to support the findings since HbA1c is a measure of glucose control over the past few months and not when the sedentary activity was actually calculated. More description on what is meant by “other reasons” and “failed to screen” are needed for Figure 1.
---

VERSION 1 – AUTHOR RESPONSE

Reviewer 1

1. Methods: “One thing that should be addressed is they excluded analyses from participants who did not have at least 4 days of sensor data, as these folks may be inherently different than those that did not capture 4 days of data. This should be presented”

This is an important comment. I have updated the flow chart to include detailed information about the exclusion based on the ActivPAL monitoring. In total 59 patients who were registered as having activity monitoring were excluded because of empty files (n=14), lost folders (n=30) or short duration (n=15). All the files were manually inspected and patients were excluded because of “short duration” if the registration lasted less than 4 days or if the sensor had been detached from the patient in periods of time, not leaving a continuous four day period.

I have inspected the characteristics of the patients in this group (n=15) compared to those included in the final analyses. There were no difference in age, function or stroke severity. Because of the low number, I have not included these numbers in the table. The other excluded (n=14+30) is because of technical issues, and is regarded as missing at random. But I do agree that it is important to describe the excluded patients, and I have therefore included the baseline characteristics of all the excluded patients (N=321) in table 1. Here we can see that the patients without activity monitoring are older, have more stroke symptoms and a lower functional level, as expected. This issue has been more clearly mentioned in the discussion on page 16.

2. Discussion: “The first two paragraphs in the discussion are essentially a re-statement of the findings in the results section and other redundancies”

In the BMJ Open author checklist #18 it says: “Summarise key results with reference to study objectives”. I have revised it, but have kept the main part, in line with the instructions.

3. Discussion: “The authors compare their findings to the existing literature, but here the writing is less strong and harder to follow their logic. This could be improved by being more a synthesis of results as compared to existing literature and an integration of how this knowledge could be incorporated into clinical care of stroke patients”

The discussion part has been edited in terms of structure and language. It is also mentioned in the manuscript that further investigations are needed, in particular studies aiming at reducing sedentary time.

Reviewer 2

1. “The discussion should a greater description of why this relationship is observed only in those with diabetes mellitus and not the whole group other than to say that the evidence in variable considering that most studies support a relationship between sedentary behaviour and diabetes risk”

I have included a molecular explanation of why this association is observed only in the patients with diabetes mellitus in the third section of the discussion part. Even though there is much evidence of the association between sedentary behaviour and glucose metabolism, the literature is characterized by varying methodology, making it hard to synthesize the results. I have therefor focused on methodology in the discussion, this being the “variable factor” and not the association per se.

2. “Including 18 and 36 months would strengthen this paper”

We agree that analysing follow-up data, and investigation the association between change in sedentary behaviour to change in HbA1c would have been interesting. However, these data were not available for this study and the objective of this study was therefore to identify an association at one point in time, i.e. at 3-month follow-up.

3. "Consider removing "in stroke patients" to update the title to "Is long bout sedentary behaviour associated with long term glucose levels three months after acute ischemic stroke? A prospective observational cohort study"

We agree. The title has been revised accordingly.

4. "Mechanism of stroke may be different in those <50 years vs 50+ years as most strokes occur after the age of 50 years. Please justify the inclusion of those <50 years. What % of the group did these individual encompass?"

The reviewer is correctly commenting that stroke mechanisms may be different <50 years and 50+. 17 (4%) of our patients were younger than 50 years at baseline. The objective of this study was to identify the association between glucose metabolism and sedentary behaviour, and stroke mechanism does not directly interfere with this association. But age is absolutely an issue when it comes to glucose metabolism, as is BMI. I have therefore adjusted for age and BMI in the analyses.

5. "Please justify the inclusion of only those able to walk >50 m? How many were excluded based upon these criteria?"

We have based the activity monitoring on position transition, arguing that the energy expenditure is higher when standing or stepping compared to in a sitting or lying position. A patient who can not walk may have a large energy expenditure, but it is not registered when he/she is not changing position (i.e. in a wheel chair). Inclusion of these patients would require a different kind of measure of energy expenditure. In our subsample (without bleeding, with valid data and blood samples: defined in the flow-chart), 3 patients were excluded because of the Barthel criteria. In the whole population at three months 91% of the patients could walk 50 meters.

6. "Where night shift workers included? Did all stroke survivors report that their normal awake time was 8am-10pm? Why include only this range when some individuals may be stay awake later and get up earlier and this should effect their sedentary time calculation."

Unfortunately, we do not have information about the work situation for the participants. However, we did a quality check by comparing the 24-hour activity data with the daytime activity data. The results showed that more than 80% of the short sedentary bouts (< 30 minutes) occurred between 8 am and 10 pm, indicating that we actually have succeeded in capturing day-time activity and excluded sleep time. Nevertheless, future research should focus on developing algorithms that are able to extract sleep-time from the 24-hour data in order to capture a greater diversity of activity patterns. This is an important issue and is therefore thoroughly outlined in a separate section of the discussion part on page 15.

7. "What % were on insulin?"

The table is updated showing the patients using insulin. In the subgroup of patients who have diabetes mellitus, 16 patients use insulin (21%). The severity of the diabetes depends on several factors, including insulin resistance and insulin deficiency. The level of insulin resistance would be relevant, as the mechanism that can explain the association found in this study is connected to

glucose consumption in the muscle. Unfortunately, we do not have laboratory results regarding this.

8. "Though the discussion attempts to put into context why HbA1c was used, measures of glucose/insulin/HOMA-IR may help to support the findings since HbA1c is a measure of glucose control over the past few months ant not when the sedentary activity was actually calculated."

In regard to using glucose/insulin/HOMA-IR, we chose HbA1c because we wanted to see the results over time. Glucose and insulin in particular have too much day- to day and diurnal variation to say anything about the habitual level. HOMA-IR would be interesting in terms of quantifying the level of insulin resistance, and I have added a section about this in the article. Unfortunately, this test was not feasible in the project.

9. "'Other reasons and failed to screen"

Reasons for "failed to screen" was amongst others: delirious patient, hearing, uncertainty about the diagnosis, multi morbid, nursing home resident, infrastructure on the ward, vacation/weekends, other studies. Details about this have not been included in the article.

VERSION 2 – REVIEW

REVIEWER	Monica Serra UT Health San Antonio, USA
REVIEW RETURNED	08-Jun-2020

GENERAL COMMENTS	Several of the reviewers' comments were addressed; however, a few minor comments remain. It was noted that a key word was type 1 diabetes. Did any subjects have type 1? Since the mechanism of disease differs greatly between T1DM and T2DM, what was the rationale for including both? The discussion states that the subjects in this study were more sedentary than their age matched peers- do you mean without stroke? I suggest adding that "on average" ischemic stroke were relatively old since some of the subjects were between the ages of 18-50. Grammatical errors are still present (i.e. BMI and age was... instead of were) Terminology is not used consistently throughout (Type 2 vs Type II diabetes).
---

VERSION 2 – AUTHOR RESPONSE

1. Comment: "It was noted that a key word was type 1 diabetes. Did any subjects have type 1? Since the mechanism of disease differs greatly between T1DM and T2DM, what was the rationale for including both?"

Answer: We strongly agree that the differences in disease mechanism between T1DM and T2DM is important to address in this study. More particularly, it is of relevance to what degree the patients have insulin resistance versus insulin deficiency. Patients with T1DM have insulin deficiency, while the relative contribution of deficiency versus resistance is less clear-cut in the T2DM-population. We

do not have information about type of DM in our population. The definition of DM was medical history, medication use or HbA1c $\geq 6.5\%$ at baseline. Details from the medical history could have been useful to identify known T1DM. At the same time, we know that up to 10% of patients believed to have T2DM, do have autoantibodies. The disease, known as latent autoimmune diabetes mellitus (LADA), can have features resembling both T1DM and T2DM, and is in the earliest stages treated as T2DM. In our data 7 of the 74 patients were treated with insulin only. These patients might represent T1DM patients, but we also know that T2DM in later stages can be treated with insulin only as well. We have therefore chosen to include all patients fulfilling the definition of diabetes mellitus at baseline without any more thorough investigations in regard to DM type. To exclude T1DM would potentially strengthen the association between sedentary behaviour and HbA1c. We have addressed this issue in the 5th section of the discussion part (the use of HOMA-IR) and in the "Strengths and limitations" section (insulin deficiency versus insulin resistance). The key words were selected from the BMJ Open database. As it is possible that patients with T1DM can have been included, both key words were selected.

2. Comment: "The discussion states that the subjects in this study were more sedentary than their age matched peers- do you mean without stroke?"

Answer: Thank you for your question enabling us to clarify this. We have added "age-matched peers from the general population" to the manuscript.

3. Comment: "I suggest adding that "on average" ischemic stroke were relatively old since some of the subjects were between the ages of 18-50."

Answer: Thank you for your suggestion. We have changed the sentence to: "The population in this study were people with ischemic stroke who on average were relatively old."

3. Comment: "Grammatical errors are still present (i.e. BMI and age was... instead of were)"

Answer: We have revised the whole manuscript with the help of a professional proofreading service for academic manuscripts and edited the grammatical errors.

4. Comment: "Terminology is not used consistently throughout (Type 2 vs Type II diabetes)."

Answer: Thank you for noticing. This has been corrected. We have chosen "diabetes type 2" to match the predefined keywords in the BMJ database.

There has been a revision of the database in regards to reasons why patients have not been included to the NorCOAST study. It turns out that 84 of the 227 registered as "other reasons" should have been registered as "failed to screen". The flow-chart has been updated.